# From the Magic Bullet to Theragnostics: Certitudes and Hypotheses, Trying to Optimize the Somatostatin Model

**DOI:** 10.3390/cancers13143474

**Published:** 2021-07-12

**Authors:** Giuseppe Danilo Di Stasio, Pasqualina Buonomano, Laura Lavinia Travaini, Chiara Maria Grana, Luigi Mansi

**Affiliations:** 1Nuclear Medicine Service, Check-Up Polydiagnostic Center, 84131 Salerno, Italy; danilo.distasio@check-up.net; 2Nuclear Medicine Service, Ios and Coleman Medicina Futura Medical Center, 80011 Acerra, Italy; pbuonomano90@gmail.com; 3Nuclear Medicine Division, European Institute of Oncology—IRCCS, 20141 Milano, Italy; laura.travaini@ieo.it (L.L.T.); chiara.grana@ieo.it (C.M.G.); 4Section Health and Development, Interuniversity Research Center for Sustainability (CIRPS), 00038 Rome, Italy

**Keywords:** theragnostics, magic bullet, somatostatin, radioiodine, DOTA, radionuclide therapy, PRRT, NETs

## Abstract

**Simple Summary:**

In oncology, the hypothetical “perfect magic bullet” should have a specific target on tumor cells which allows one to target only the tumor, in the absence of uptake in normal and/or non-neoplastic cells. Theragnostics is a strategy that strictly combines diagnosis and therapy, which creates the conditions for an “a priori” definition of an effective therapeutic effect. The most complete theragnostic and “magic bullet” experiences in clinical practice are those associated with radioiodine and somatostatin model. In this paper, we analyze whether it could be possible to improve present clinical results, further extending the survival of a wider number of patients, expanding the recruitment criteria to other types of pathology, and improving the quality of life. The ultimate goal is to transform the theragnostic strategy based on the somatostatin model into a curative therapy in the highest possible number of patients.

**Abstract:**

The first “theragnostic model”, that of radioiodine, was first applied both in diagnosis and therapy in the 1940s. Since then, many other theragnostic models have been introduced into clinical practice. To bring about the closest pharmacokinetic connection, the radiocompound used for diagnosis and therapy should be the same, although at present this is rarely applicable. Today, a widely applied and effective model is also the “DOTA-Ga-68/Lu-177”, used with success in neuroendocrine tumors (NET). In this paper, we analyze the necessary steps from the in vitro evaluation of a target to the choice of radionuclide and chelate for therapy up to in vivo transition and clinical application of most employed radiocompounds used for theragnostic purposes. Possible future applications and strategies of theragnostic models are also highlighted.

## 1. Introduction

In 1900 the Nobel laureate Paul Ehrlich coined the term *Zauberkugel* (magic bullet) to define a hypothetical agent able to kill microbes, while safeguarding their human carriers. The necessary premise was the presence of a specific target on the pathogen agent, but not on human cells [1].

Since then, the concept of magic bullet became a goal to be reached, mainly in oncology, giving support to the research of a large series of molecules that could possibly interact with a specific target and/or show an uptake mechanism present or highly increased in the malignant neoplasm, but not in normal or non-neoplastic cells. In this way, it would have been possible to obtain a pathognomonic diagnosis of cancer, having the opportunity to acquire tools for an effective therapy against the tumor, without significant side effects.

This concept has been given particular relevance in the context of nuclear medicine, whose specialists can trace in vivo whole-body distribution of a target and/or of a concentrating function, using radiotracers mimicking biomolecules involved in pathophysiological alterations, confidently typical of malignancy. The history of the discipline has been marked by eras of high hopes in which it was believed possible to produce and utilize a magic bullet in routine practice. However, none of the many strategies clinically applied have fully accomplished the goal. Such strategies include those using the so-called “tumor-seeking indicators”, such as Ga-67 citrate and Tl-201 chloride, radiolabeled monoclonal antibodies, used not only for diagnostic purposes but also for radioimmunotherapy, and many metabolic and/or molecular radiotracers [2,3,4,5]. In recent years, a revolution has shifted the axis of rotation of the diagnostic universe from structural to functional imaging, because of advantages not only in diagnosis, but also in prognostic stratification and in a better interaction with therapy. The advent and the widespread diffusion of hybrid imaging has further improved diagnostic accuracy. Nevertheless, a pathognomonic cancer diagnosis and a consequent effective therapy, without side effects, still remains a chimera in the vast majority of cases.

In this context, the optimistic vision supporting researchers and industry in continuing to seek out new magic bullets was stimulated by the effectiveness of the first “theragnostic model”, that of radioiodine, already applied, as I-131 sodium iodide, both in diagnosis and therapy since the 1940s [6,7,8]. The burgeoning interest was also strongly encouraged by the incredible development of molecular biology and the consequent birth of precision and tailored medicine [9,10].

By “theragnostics” we mean a strategy, intimately intertwining diagnosis and therapy, which creates the conditions for an “a priori” definition of an effective therapeutic effect. Broadly, this concept includes a wider series of associated approaches, such as diagnostic imaging to evaluate therapeutic response or to guide radiotherapy, or the bio-molecular detection on histological samples of the expressed oncogenes, to guide targeted therapies. A more restricted meaning of theragnostics may be limited to the availability of a molecule and/or a container object that can be labeled with a different tag, without significantly changing its in vivo kinetics. This tool must be able to be visualized from the outside using the diagnostic molecular form and, changing the label, to direct a toxic agent to the targeted cells, when a therapeutic effect is required. These methods, primarily developed in the field of nuclear medicine, have more recently been activated also in radiology, mainly using nanotechnologies, allowing the application of the same carrier either as contrast media for diagnosis or associated with a toxic agent for therapy [11,12].

As previously reported, the best theragnostic model presently utilized in clinical practice still remains the first one based on radioiodine (I-123 for diagnosis and I-131 sodium iodide for therapy), which has the advantage of its almost specific uptake in thyroid tissue. Therefore, following a consolidated procedure, developed and improved over the decades, an effective curative therapy may be obtained in the majority of patients affected with differentiated thyroid carcinoma. This can also happen in the presence of metastases, the iodide symporter still remaining active in well-differentiated neoplastic cells, also at the level of secondary lesions. 

It should be pointed out that a major advantage of radioiodine is determined by its molecular form, sodium iodide, a small and simple molecule, which remains very stable after labelling with I-123 or I-131. Furthermore, the pharmacokinetics of radioiodine is almost overlapping with respect to the native stable I-127 iodide.

More recently with respect to radioiodine, many other theragnostic models have been increasingly introduced into clinical practice, particularly in recent decades. Favorable results have been obtained for example, in neuroblastoma, paraganglioma, medullary thyroid carcinoma, and pheocromoblastoma, using *meta*-iodobenzylguanidine (MIBG), labelled with I-123 or I-131, as well as osteotropic agents (radiolabeled with radionuclides emitting gamma-rays or positrons for diagnostic use, with beta or alpha radio-emitters for therapy), monoclonal antibodies, applied in the field of many neoplastic diseases using different radionuclides either for diagnosis or therapy, and the Prostate-Specific Membrane Antigen (PSMA), a theragnostic model in prostate cancer, mainly utilizing Ga-68 and F-18 for diagnosis and Lu-177 for therapy [13].

In this paper we wish to analyze what may be considered the most complete theragnostic experience, that associated with the somatostatin model. Native somatostatin is unusable in vivo for therapeutic purposes, because of too-rapid pharmacokinetics. This greatly spurred researchers on to synthesize somatostatin analogues (SSA), as they have a slower wash out. The SSA octreotide was firstly used in pituitary adenomas. Since then, a full series of radiocompounds, labelled with gamma emitters, positrons, beta, and alpha emitters, has been obtained to respond to all the diagnostic and therapeutic queries in an almost “theoretically perfect” theragnostic strategy, which has been widely applied in neuroendocrine tumors (NETs) [14]. Reality of course differs from the theoretical world, and we will discuss here whether it is possible to reach a further goal, namely that of optimizing a strategy based on radiolabeled SSA as a main component of a therapeutic strategy for all NET patients.

Somatostatin and SSA are peptides, i.e., short chains of amino acids, linked by peptide bonds, which do not exceed 50 elements, showing a molecular weight of less than 5000 Daltons. SSAs are easily synthesized and may be chemically modified without particular problems. Using chelators, it is possible to produce a large series of stable radiocompounds, labelled with different radionuclides, exploitable either for diagnosis or therapy, with a similar pharmacokinetic creating the premises for an effective theragnostic model. When injected in vivo, radiopeptides are characterized by an excellent permeability and an elevated first pass extraction that, in the presence of high-affinity receptor binding, is associated with a rapid clearance from the blood. Being non-toxic at the administered dose, and showing a very low or absent antigenicity, in vivo administration only rarely creates minimal side effects. Their clinical application is favored by the advantageous distribution of somatostatin receptors (SSR), highly expressed in the majority of differentiated NETs, being scarcely present in normal and non-neoplastic cells. A particular exception, that can however allow interesting clinical applications, is represented by the increased expression of SSR on the membrane of reactive cells, such as macrophages, endothelial cells, and lymphocytes, when activated.

To better understand whether it is possible to further improve the clinical usefulness of radiolabeled SSA in NETs, we will start by analyzing all the steps which, from the in vitro evidence of SSR on neuroendocrine neoplastic cells, arrive to a fully effective therapy.

At present, Peptide Receptor Radionuclide Therapy (PRRT) is already clinically feasible in the treatment of somatostatin receptor-positive gastro-entero-pancreatic neuroendocrine tumors (GEP-NETs), using Lutetium (Lu-177) oxodotreotide (Lutathera^©^), approved by the FDA, EMA, and national agencies. Nevertheless, here we wish to explore whether it could be possible to improve present clinical results, further extending the survival of a wider number of patients, expanding the recruitment criteria to other types of pathology, and improving the quality of life in all the treated patients. The ultimate goal is to transform the theragnostic strategy based on the somatostatin model into a curative therapy in the highest possible number of patients.

## 2. Differences between In Vitro Experimental Models and Clinical Practice

The “perfect magic bullet”, that makes Paul Ehrilch’s dream in oncology become reality, should determine in vivo a very high tumor to non-tumor (T/NT) ratio, due to its specific ability to target the tumor, in the absence of uptake in normal and/or non-neoplastic cells. This specific target must be present on all the neoplastic cells, both in the primary tumor and in metastases, and be reachable in all the lesions, due to the absence of pathophysiological barriers and/or of a biological state that could otherwise prevent or limit the bullet’s concentration. When in situ, in the therapeutic phase, this agent has to determine an irreversible tumor cytotoxicity, in the absence of relevant side effects, both at local and general level. With respect to pharmacokinetics and use in therapy, it is important that a standard procedure be in place, allowing a repeatable behavior after each administration, with a distribution determined by mechanisms that are identical in all the patients undergoing this strategy in each of the therapeutic steps.

Nevertheless, it must be emphasized that using radionuclide therapy, a cytotoxic effect, may be obtained not only through a direct action on targeted cells, but also because of indirect effects such as those resulting locally from a sequela in the stroma or in the environment surrounding the concentration sites. A possible detrimental consequence on non-targeted cells may be also obtained at a distance from the concentrating site, when the activation of a general response, such as the immunological response, is able to determine a toxic effect independently of the presence of a highly specific mechanism of concentration. Therefore, it is possible, using a targeted therapy, to obtain a therapeutic effect also on neoplastic cells not expressing the target. This means that in vitro determination of a specific target, such as SSR in GEP-NETs, must be the first step to activate the theragnostic path. Nevertheless, the in vitro expression of SSR (defined on a pathological sample or considered highly probable in the presence of a suspicion of GEP-NET or of a similar disease), although necessary to start the whole procedure, is not sufficient to guarantee a final curative effect.

It will not suffice merely to find a promising magic target in preclinical research if we want to have the certainty of an effective in vivo therapeutic response. Similarly, it will not suffice to demonstrate the presence of a “theoretical magic target” at biopsy and/or primary tumor level to be confident of a fully eradicative strategy, capable of destroying all the neoplastic lesions, including metastases, while safeguarding the human carrier of the disease.

Furthermore, while the theragnostic model utilized in the treatment of thyroid carcinoma, using radioiodine as sodium iodide, is favored by the availability of a chemical form not significantly modified in its biological behavior by radiochemistry, the radiolabeling of SSA does create certain discrepancies with respect to the “cold” molecule. Moreover, in vivo pharmacokinetic differences are observed not only with respect to the unlabeled molecule, but also among the different radiocompounds used for diagnosis or therapy. In this sense, to arrive to the best performance in PRRT, a more complex and rigorous approach is warranted with respect to the radioiodine approach. For each specific purpose, the choice not only of target and bullet, but also of the best radionuclide and radiocompound, needs to be carefully considered. In particular, to achieve the greatest results in the whole path, it can be important to determine, more than the outstanding diagnostic or therapeutic agents considered separately, the theragnostic couple that most resemble each other. This is important because, when administering therapeutic doses, our goal has to be reached not only considering the tumor/background ratio, but also the uptake at the level of the critical organs, such as kidneys. A dosimetric evaluation, performed either in a pre-therapeutic stage or even during therapy, can significantly improve therapeutic efficacy and quality of life. It can be better obtained when diagnostic and therapeutic tools present a similar pharmacokinetic and bio-distribution, allowing a correct a priori prevision of a fully eradicative action.

To understand all these points, we will then individually analyze the various stages that form the basis of the best final therapeutic effect.

### 2.1. In Vitro Choice of the Target (and of Bullets)

The first step in identifying the perfect target is in vitro. Starting from a theoretical premise suggesting the choice of a characterizing indicator of neoplastic disease, an extensive evaluation of the presence of this target on neoplastic and normal cells, from histological samples, is then performed. Depending on its possible use within a theragnostic system, it is important that this marker is stable and accessible and therefore expressed on the cell membrane and hopefully of all the neoplastic cells, both in the primary and secondary lesions.

Going back to the somatostatin model, very favorable results have been obtained in vitro, strongly supporting a research goal of identifying bullets directed against somatostatin receptors (SSR). The kinetics of native somatostatin are too rapid to be feasible in the clinical practice. Therefore, research was directed to the synthesis of SSA, showing a slower washout after in vivo administration. Of the five different SSR subtypes characterized, octreotide, the first clinically available SSA, was mainly interacting with SSR2 and SSR5. Luckily, these two receptors are hyperexpressed in the large majority of GEP-NETs, and in many other NETs, at the level of the primary tumor, metastases, and recurrence. Conversely, they are scarcely expressed in normal cells and in the majority of the other neoplasms. An almost similar distribution has been observed for many of the other SSA (either “cold” or radiolabeled) further introduced into clinical practice, both for diagnosis and therapy. At present, a pan-receptor SSA, having the capability to interact with all the SSR subtypes, is yet to be commercialized [15,16]. Therefore, a clinically useful theragnostic model based on radiolabeled SSA, conditioned from the individual receptor binding profile of somatostatin, has been mainly applied in GEP-NETs. In this group, insulinoma is an exception, because of an unfavorable SSR2 and SSR5’s expression. Moreover, it has been demonstrated that SSRs are associated with the differentiated neoplasm and they can be lost in the de-differentiation process, as observed in the follow up of the forms with the worst prognosis. A lesser expression can be seen in some of the neoplastic lesions in the most malignant cases, limiting an effective therapy also in these patients.

With SSR expression in the majority of normal cells and of other neoplasms being absent or low, an increased expression is observed in reactive cells, such as macrophages, lymphocytes, and endothelial cells, when activated. This may become a useful issue for an in vivo application, it being possible to obtain a local in vivo uptake of radiolabeled SSA in tumors which are non-expressing of SSR, such as non-small cell lung cancer (NSCLC), or high-grade glioma in the presence of a local infiltration by activated reactive cells. Furthermore, the high expression on these cells may become a favorable pathophysiological premise for a clinical use to both define disease activity and guide therapeutic strategies in benign chronic diseases, such as exophthalmos in Graves’ disease or rheumatoid arthritis.

Autoradiography and other in vitro preclinical techniques may also allow the quantification of receptor’s density and permit the acquisition of much data useful to understand basic characteristics of NETs, relevant to support the clinical use of the theragnostic model. In this context, the receptor activation, which operates on tumoral growth (and on hormonal hyper-secretion), may be observed and understood. Similarly, the hetero-dimerization of SSR subtypes and their internalization may be demonstrated. With respect to the somatostatin model, a possible neo-angiogenic effect and the ability to activate an immunological response have also been demonstrated in vitro as important pathophysiological premises for a more effective in vivo therapeutic effect [14].

### 2.2. The Need for In Vivo Research

In vitro research and/or the analysis on a pathological sample is a mandatory step in defining the feasibility of a theragnostic model, but it is not sufficient to predict the in vivo therapeutic efficacy of the identified bullet [17]. In fact, in each individual with an NET, many questions cannot be answered such as: (1) intra- and inter-lesion variability: is the target expressed in all the neoplastic sites and extensively in each single lesion? (2) temporal variability: may an SSR expression disappear in the follow up, because of the so-called “escape phenomenon”? (3) pathophysiological conditions: may the bullet reach in vivo all the neoplastic lesions, despite the possible presence of altered flow, necrosis, and biological barriers? Furthermore, in vitro research can only evaluate single and/or local components of the human being, not having the ability to analyze complex phenomena, such as the immunological response or neoangiogenesis. Moreover, everything that is biodynamic, with main reference to pharmacokinetics and bio-distribution, i.e., parameters which can strongly influence a therapeutic effect, cannot be understood without an in vivo evaluation. From the above, it is clear that integrating animal studies into preclinical research is mandatory. This need assumes even greater value in the field of theragnostics, because the radiolabeling of molecules, using diverse radionuclides and doses, produces different pharmacokinetic behaviors. The relevance of this evaluation is further increased by the emerging evidence of in vivo mechanisms associated with radionuclide therapy, which are not only limited to the so-called crossfire effect. This action, previously considered as almost exclusive for the therapeutic efficacy of a radiocompound, is mainly conditioned by the energy spectrum of the labelling radionuclide, which allows the possibility to destroy also non-targeted neoplastic cells, but only in the range of the ionizing radiation, also considering the different tissue’s attenuation. More recently, the possible presence of further mechanisms, allowing the killing of non-targeted neoplastic cells even beyond the range of radiations emitted, has been identified. In particular, the bystander effect may determine the death of neoplastic non-targeted cells, near to those concentrating the radiocompound, because of the transmission by contiguity or through the elimination in the interstitial space of toxic agents, such as free radicals, catabolites, and inflammatory factors, consequent to the destruction of the targeted cells caused by radioactivity. A great interest is also growing in a deeper understanding of the abscopal effect, trying to explain the possible remote therapeutic effect also in non-targeted metastatic cells. This phenomenon seems to be determined by local activation, in the site of the specific concentration of the radiocompound used for radionuclide therapy, of a general humoral response directed against determinants present either on the primary tumor and on metastases, even if less differentiated. All this information can only be achieved in vivo. This is why primary relevance is given to preclinical research in experimental animals before applying a theragnostic model in humans.

### 2.3. Need for a “Well-Functioning” Theragnostic Model In Vivo

In vitro research is mandatory, but not sufficient to individuate a theragnostic model that is well-functioning in vivo.

With respect to the target’s expression, four different cell populations are identifiable in vitro: neoplastic or normal cells, either expressing or not expressing the target. Conversely, the achievement of the in vivo T/NT after the administration of radiolabeled SSA may be conditioned by further parameters, determining uptake both in cells or in extracellular components, which may change the theoretical distribution of the radioactivity and, as consequence, the therapeutic efficacy. At first, there is no possibility of a specific targeted uptake in the presence of a pathophysiological barrier. For example, cerebral metastases, also if hyper-expressing SSR, cannot be reached and destroyed by radiolabeled SSA, if the radiotracer is unable to pass through the blood–brain barrier (BBB). It has to be specified, however, that with gliomas or secondary lesions to the brain, BBB is most often broken too, with tracers, as well as other compounds or drugs, that can get through the barrier and reach neoplastic cells. Furthermore, an in vivo uptake of radiolabeled SSA may be observed in vivo at level of a tumor, also if derived by in vitro SSR-negative neoplastic cells, when in the presence of a local concentration of activated reactive cells highly expressing SSR. This behavior has been observed, for example, in NSCLC [18]. Moreover, with the ability to produce in vivo many new radiochemical forms with respect to the one injected, such as catabolites, free radionuclide, radiocomplexes, etc., radioactive uptake may be also consequent upon non-specific mechanisms. All these conditions may affect the bio-distribution of a “radioactivity”, now intended as sum of many radiochemical forms, conditioned by pathophysiological phenomena partially independent of the specific interaction SSA-SSR. In this way, radioactive uptake may be observed at level of active scars, in the presence of an increased permeability, altered flow, increased blood pool, as may happen in inflammation, or may be conditioned by necrosis, and so on. With the therapeutic efficacy being conditioned by the T/NT ratio, a major compartment that has to be carefully evaluated, before starting with a PRRT, is the excretory system. In particular, the renal uptake may represent a limiting factor in the administration of effective doses, in cases of a high regional radioactivity, and many studies have been performed to find the way to reduce radioactive concentration in kidneys and in the emunctory systems [19].

It has to be taken into account that the radioactive uptake observed in active lymphocytes, monocytes, and epithelioid cells is also “specific”. Nevertheless, while a similar pharmacokinetics may be observed in normal or reactive cells, when they are stationary, a faster wash-out may be seen at the level of the “hot” peri-tumoral and/or intra-tumoral region in negative SSR tumors, such as NSCLC, being the uptake dependent on “circulating” cells over-expressing SSR. A non-neoplastic “specific” concentration may also be associated with active epithelioid cells, which can be present at level of neo-angiogenic vessels. The different kinetics observed in vivo in SSR positive tumors, such as SCLC and carcinoids, and negative SSR neoplasms, such as NSCLC, also showed a high Octreoscan© uptake because of the presence of peri-tumoral SSR positive reactive cells, which prompted Briganti et al. to propose a two-time scan (at 4 and 24 h) to differentiate the two tumors, typing them in vivo [20]. In particular, while an increased uptake was observed at 24 h in SCLC (or carcinoids), a stationary or decreased concentration was seen at the level of NSCLC, because of an uptake mainly dependent on mobile cells. A further support to the explanation of a faster wash-out dependent on SSR positive reactive cells has been given by Cuccurullo et al., who demonstrated, using 111In-pentetreotide in patients with Graves’ disease and active exophthalmos, a faster wash-out at the level of the orbits, in the presence of circulating lymphocytes and macrophages, with respect to the thyroid in which the uptake was associated to stationary cells [21].

The presence of multi-parametric pharmacokinetics must be taken in account to acquire a better dosimetric information before therapeutic administration. In particular, it is not only important to define the temporal variation at the level of the tumor and background, but also the dynamic behavior at the level of the kidneys, to calculate the total dose not only on the basis of the renal uptake, but also of the radioactivity persistence and wash out time, also in the presence of associated treatments, such as the amino acid infusion. Thanks to this type of study, the capability to determine a renal protection has been achieved, allowing the possible recruitment of a wider number of patients for PRRT. Furthermore, this information may entail the administration of increased therapeutic doses, given the reduction in renal damage [19,22].

Interestingly, the specific uptake at the level of reactive cells may also increase the indications of radiolabeled SSA. For example, Cuccurullo proposed Octreoscan© for radioguided surgery and intra-surgical staging of resectable pulmonary neoplasms, including not only SSR positive carcinoids, but also NSCLC, showing a high in vivo uptake dependent on peri-tumoral reactive cells [18]. On these pathophysiological premises, a “theragnostic” approach could be (eventually) utilized also, using as therapeutic agents cold SSA, also in non-malignant pathologies, such as pituitary adenomas (when SSR positives, as in GH-secreting adenomas) or active chronic diseases (therefore showing high uptake of reactive SSR positive cells), as may happen in Graves’ exophthalmos, rheumatoid arthritis, Crohn’s disease, and so on.

## 3. Radiocompounds for a Theragnostic Model

In a theragnostic perfect model, the radiocompound used for diagnosis should be the same as that used for therapy, to realize the closest pharmacokinetic connection. Nevertheless, this condition is at present the exception, being actually rarely applicable, as may happen for Lu-177 radiocompounds and, partially, for radioiodine; for the latter, to reduce the dosimetric charge in the diagnostic phase, I-123 is preferred over I-131, used only for therapeutic purposes (either as sodium iodide or MIBG). Although some kinetic differences may be observed, because of the different energy spectrum (for radioiodine) and/or dose, the bio-distribution of the radiotracers proposed for diagnosis or therapy is substantially overlapping.

Despite being a radionuclide particularly suitable for the radiosynthesis of molecules, especially for proteins or peptides, radioiodine (as I-123, I-124, I-125, I-131), and radio-iodinated tracers are negatively affected by the in vivo de-iodination, determining an unjustified radiation dose to the thyroid, salivary glands, and bowel. Although a blockage of the thyroid uptake (the greatest limiting factor) may be obtained, it has been preferred to synthetize more stable radiocompounds for therapeutic purposes. By analogy with radioiodine, radio-fluorinated compounds (affected by an in vivo de-fluorination), although widely used for diagnosis, have been rarely proposed for a theragnostic model in connection with therapeutic agents. Conversely, a greater interest has been devoted to radio-chelates, which guarantee a radiochemical stability for many hours, allowing a radiochemical synthesis of diagnostic and therapeutic agents, using the same molecule and different radionuclides. Furthermore, chelated radio-molecules are highly stable in vivo for many hours, ensuring a substantial kinetic overlap between the chelated radiotracers used in diagnostics and therapy, regardless of the differences in radionuclide and dose.

In the context of the radio-chelate family, the somatostatin model has been the first and is still the best example on how this group of radiocompounds may acquire a primary position in the theragnostic universe, demonstrating that is possible to label the same molecule with gamma, beta, or alpha emitters.

In this way, an a priori diagnostic evaluation, which may also include a pre-therapeutic dosimetric information, may be acquired as a precondition to recruit patients for PRRT, having a high probability of an effective therapy, because of strong kinetic similarities between the diagnostic and therapeutic radiotracer.

Nevertheless, also considering the different energy spectrum and half time of the radionuclides utilized, it is however important to verify the presence of possible differences connected with the radiochemical form, problems of radiolysis that may increase at higher doses, a greater ponderal amount of the molecular carrier, and a change in specificity determined by the different radiochemical production. Furthermore, it has to be remembered that the uptake and the consequent pharmacokinetic of a radiotracer may be modified by a previous administration of radioactive compounds competing for the same uptake mechanism, interfering drugs, physiological, para-physiological, and pathological variations present at the time of a different temporal administration for the diagnostic and therapeutic radiotracer.

In the definition of a “well-functioning theragnostic agent”, it is particularly important to remember how the distribution of radiolabeled SSA, used for diagnosis or therapy, may be conditioned by the previous administration of drugs, with main reference to the administration of cold somatostatin analogues. As a consequence, a protocol concerning PRRT has to be carefully applied, also taking into account a precise temporal schedule between diagnosis and therapeutic administration, considering all these issues.

On the basis of the information reported above, we can identify three different options to determine a well-functioning connection between the diagnostic and the therapeutic phase in NETs using radiolabeled SSA: (1) independent choice of the radiotracers used for diagnosis and therapy. Using this approach, to define the presence of SSR as targets, also radio-peptides labelled with Tc-99m or F-18, showing a different bio-distribution with respect to the radio-agents used in PRRT, may be included, bearing in mind that there is no direct counterpart on the therapeutic versant; nevertheless, the diagnostic radiotracers are able to correctly recruit patients for PRRT; (2) radiotracers having the same and/or a highly similar radiochemical structure, with main reference to the carrier molecule and to the chelating agent. This is the most diffuse operative “theragnostic model”, based on Ga-68 (or more rarely In-111, in the absence of PET/CT), as radionuclide for diagnosis, and Y-90 or Lu-177 as radio-agents for therapy. The same combination has also been proposed in the case of the possible use of alpha emitters at a therapeutic level; (3) use of the same radionuclide, at different doses, either for diagnosis and therapy, as may be done with radiotracers labelled with Lu-177, emitting either beta or gamma radiations. In this group, we could also include In-111, having also a therapeutic effect through Auger electron emission. Having proposed and applied radio-peptides labelled with In-111 in the initial experiences of PRRT, these have been completely abandoned in radionuclide therapy, due to their low efficacy, having been almost totally replaced in the diagnostic phase by the better-performing DOTA peptides labelled with Ga-68 [23,24], or other radionuclides such as Y-86 [25], Cu-64, and Co-55 [26].

From a theoretical point of view, the best theragnostic model should be by analogy with radio-iodine, which uses the same molecule at different doses. This application could be affected by negative interactions, as may happen for the so-called stunning effect for I-131 sodium iodide, reducing its therapeutic beneficial efficacy, if the diagnostic dose has been administered shortly before. Similarly, from a theoretical point of view, Ga-68 could be considered impractical for a theragnostic model because of a too-short half-life, which prevents investigators from evaluating the kinetics directly and, as consequence, the dosimetry of therapeutic radionuclides characterized by a longer decay. Nevertheless, in vivo radiochemical stability makes this minor limitation irrelevant.

Therefore, in the practical world, the diagnostic choice is based on the preference for a choice privileging the best procedure to recruit patients for PRRT, using radio-chelate allowing also a PRRT. In this sense, Ga-68 radiotracers are preferred with respect to those labelled with Lu-177, when administered at diagnostic doses, because of a higher sensitivity and accuracy, in the presence of a lower dosimetry. Conversely, in a comparison limited to Lu-177 and Y-90, the Lu-177 energy spectrum, characterized by the presence of a gamma emission allowing SPECT studies of a sufficient quality can be considered an accessory advantage with respect to Y-90, permitting the achievement of low-quality images only through the bremsstrahlung effect (Table 1).

For all the reasons reported above, if we are to define the best choice, it is important to demonstrate an in vivo radiochemical stability, which can maintain the original administered form for many hours in the presence of a scarce catabolism or of a very low liberation of the free radionuclide. Furthermore, it is important to demonstrate a similar pharmacokinetics between the diagnostic and the therapeutic radiocompound, considering the different doses.

For many years, numerous studies have already demonstrated an almost overlapping biological behavior between SSAs radiolabeled with different radionuclides. Substantially overlapping pharmacokinetics have been observed using DOTATOC labelled with 67Ga/68Ga [27] or 111In [28], or between 86Y-DOTA octreotide and 111In-pentetreotide [29]. Conversely, differences have been seen when comparing 99mTc or 18F radio-peptides with respect to the corresponding 90Y or Lu-177 radio-chelates used for PRRT [30,31].

### 3.1. Choice of the Radio-Chelate

With respect to all the chemical components included in the theragnostic model “Somatostatin”, the chelating agent is also a central constituent, which may slightly affect the perfect implementation of the theoretical model because of slight pharmacokinetic differences where a diverse chelate is used for diagnosis or therapy. Slight variations should also be due to a possible detrimental effect on the peptide’s function, and therefore on the affinity with SSR, more frequently determined by the higher radioactive dose associated with the therapeutic agent.

When using a radiotracer, it is well known that while the radionuclide is the part allowing the in vivo detection, it is the carrier which determines the bio-distribution. It is very important that a high receptor affinity of the SSA be maintained after labelling. This is because a different affinity of radiolabeled SSA for SSR may be due not only to the high inter-patient variability, but also, in the same subject, to a different receptor density on individual neoplastic cells. This condition can create a significant discrepancy between multiple lesions in the same subject, with high radioactive concentrations in some lesions, the uptake in other localizations being absent due to the absence of receptors interacting with the utilized radiolabeled SSA. This situation may mainly occur in follow up, because of the appearance of de-differentiated cells, which may also determine the loss of a previous lesion’s uptake, due to the so-called escape phenomenon. For these reasons, it is important to have confidence in the possible differential behavior between the diagnostic and the corresponding therapeutic radiotracer, knowing that different radio-chelates may have a diverse affinity for SSR. In the same way, it being possible that the diagnostic radiolabeled SSA has a different renal concentration with respect to the therapeutic agent, one must be very careful in quantitatively predicting a priori the renal damage due to PRRT.

To demonstrate how a different distribution may be determined by the chelate, and more in general by the labelling procedure, we report studies by Forrer et al. [28]. To evaluate a possible different efficacy of Lu-177-DOTATATE versus 90Y-DOTATOC, both used in PRRT, and they compared bio-distribution in patients with metastatic NETs using 111In-DOTATATE or 111In-DOTATOC. Using this approach, based on the administration at the same dose of In-111, a surrogate for both Lu-177 and 90Y, the differential bio-distribution may be only due to the different radio-chelate. On the basis of this experience, evidence of a different dosimetry (and therefore variations in the therapeutic efficacy) was demonstrated in the same patient, although non-univocal results in favor of a specific radio-chelate have been obtained using different experimental models. For example, while De Jong et al. [32] demonstrated a different performance in rats bearing the CA 20948 tumor, Froidevaux et al. [33] showed a similar bio-distribution of 111In-DOTATOC and 90Y-DOTATOC in the AR4-2J bearing mouse model. As previously noted, it has also been shown that a dissimilar renal adsorbed dose after amino acid co-infusion may be obtained using different radio-chelating methods, which can be based on a neutral charge, as for 111In-DOTATATE, or a positive overall charge, as for 111In-DOTATOC [28]. Further evidence in functional modifications determined by radiolabeling have been proven by Kwekkenboom, individuating a fourfold greater tumor concentration for Lu-177-DOTATATE with respect to 111InDTPA octreotide [34]. In another study, Reubi showed in vitro that 90Y-DOTATOC has a sevenfold lower binding affinity for SSR2 with respect to 90Y-DOTATATE, supporting the hypothesis of better results achievable in vivo using DOTATATE for diagnostic or therapeutic purposes [35]. Conversely, the in vivo experience reported by Forrer suggested almost similar results, with small advantages for 111In/90Y-DOTATOC, showing a better detection of liver metastases in the presence of a lower hepatic adsorbed dose, and a more favorable dosimetry at the level of kidneys. Conversely, some animal studies demonstrated a better bio-distribution for the TATE-radio-chelates [28,36].

All the data above clearly demonstrate how important it could be to personalize the theragnostic model in each individual patient undergoing PRRT. It is however evident how this strategy is very difficult, if not impossible, in daily clinical routine. Therefore, a pivotal event has been the exploitation of NETTER-1 phase III multicenter trial that confirmed the clinical efficacy of Lu-177-oxodotreotide (Lutathera©) in 229 patients with advanced, progressive, SSR-positive midgut neuroendocrine tumors, compared to best supportive care (i.e., octreotide LAR alone at a dose of 60mg every four weeks). As their results demonstrated, treatment with Lu-177-oxodotreotide provided higher response rates with significantly longer progression-free survival and overall survival (Figure 1) [37].

### 3.2. Choice of the Radionuclide for Therapy

As reported above, for the diagnostic radiotracer to be used in the theragnostic model of NETs, the choice of the radionuclide may be influenced by the need for the widest similarity in pharmacokinetics with respect to the corresponding therapeutic agent used in PRRT. In this sense, although a radio-fluorinated radiocompound could give better results with respect to one radiolabeled with Ga-68, the choice tends to favor radio-chelates, because in this way the therapeutic radionuclides may utilize the same radiochemical synthetic process, i.e., the chelation, resulting in an almost identical pharmacokinetic.

When we have to separately consider the choice of the radionuclides to be used in PRRT, starting from the hypothesis that the molecules to which they are linked are substantially the same, resulting a similar bio-distribution, many issues need to be taken into consideration.

At first, it is important to predict with the greatest possible approximation, individually for each patient as evaluated in the preliminary diagnostic phase, the bio-distribution of the therapeutic agent. It must be remembered that the radioactive delivery and the subsequent pharmacokinetics is caused not only by the receptor binding to the neoplastic cells, but also by all the in vivo connections associated with specific and non-specific radiochemical forms and by an undetermined concentration, independent of the specific link. This situation produces generally negative consequences, as the high concentration at the level of the renal emunctory, representing a limiting factor for PRRT [19], or an increased activity at the level of active scars or inflammatory sites, mainly in the presence of an altered permeability. Conversely, a positive effect may be also obtained when a specific and/or non-specific concentration may act on tumor vessels or stroma, determining a complementary therapeutic effect, for example, through the production of fibrosis or necrosis. This activity, working as a local radiotherapy, may be associated with a specific uptake of radiolabeled SSA on activated reactive cells, as lymphocytes, macrophages, endothelial cells surrounding or included in the neoplasm, being allocated in the stroma. Similarly, it could be also connected with a non-specific and/or undetermined uptake.

However, the most important issue to be considered in the choice of therapeutic radiolabeled SSAs usable for PRRT is connected with their physical characteristics, i.e., energy spectrum, type of radiations, and physical decay. With respect to the latter, together with the physical half-life, it is also important to consider the biological wash-out, both defining for each individual the effective half-life, also conditioned by the emunctory system functional state.

Historically, there have been two main radionuclides proposed, Y-90 and Lu-177, both emitting beta radiations, and these are still the most widespread for PRRT. In the beginning, many years ago, also the gamma emitter In-111 (at high dosage) was utilized for therapeutic purposes, because of the Auger electron production. These high LET electrons have an in vivo range varying from 10^−3^ mm to 10^−6^ mm. Their therapeutic action requires therefore an internalization by the cell, to have the capability to determine a DNA disruption. This condition is accompanied by a dosimetric advantage, the detrimental effect on non-concentrating normal cells adjacent to the site of concentration being low or absent. Nevertheless, a major negative limitation is due to the need for a very wide and high expression of SSR on all the neoplastic cells, the therapeutic action of radiolabeled SSA only being guaranteed in cases of internalization. As a consequence, although a relatively favorable radiation exposure for the patient, PRRT with In-111 has been abandoned because of the high probability of failure in achieving a complete therapeutic response.

More recently, other therapeutic radionuclides have been considered as therapeutic agents, with a growing interest for alpha emitters or other pairs of theragnostic isotopes [38].

For many years, with respect to corpuscular radiations usable in radionuclide therapy, the use in humans has been only limited to the beta radiations. In fact, alpha rays, characterized by a very high ionizing power and a low penetration, being stopped in vivo in a range less than 100 µm (by a sheet of paper), were considered ineffective, in the presence of a tumor showing a non-homogeneous uptake, and very dangerous in case of a high concentration in normal tissues, with main reference to the bone marrow.

In this sense, PRRT only came into play through the recruitment of the beta emitters Y-90 and Lu-177, both having a chemical structure suitable for a chelation in line with that of Ga-68 or In-111, i.e., using the same chelating agents. In this way an effective theragnostic strategy may be applied in NETs.

As it can be demonstrated from Table 2, Lu-177 has a mean beta energy of 133 KeV (497 as maximum) versus a mean energy of 933 KeV (max 2284) for Y-90. Consequently, the radiation range in water is 1.9 mm for Lu-177 and 11.8 mm for Y-90. Because of this different extension of the effect, Y-90 should work better for non-homogeneous and/or larger neoplastic lesions, the use of Lu-177 being preferable in the presence of a wide and high hyper-expression of SSR on multiple little lesions, i.e., in cases of well-differentiated disease, which is the most frequent population with a good response after PRRT [39].

To define dosimetry, it is also important to report that Y-90 has a shorter HL (64 h vs 6.71 days), and this difference may produce a lower dose to the neoplastic lesions, because of the slow washout from the site of a radioactive concentration. Furthermore, the different decay is not relevant in producing a more favorable tumor/background ratio, this being that the risk of renal damage is higher when using Y-90. Likewise, because of a favorable gamma emission allowing the attainment of SPECT/CT images of good quality also in the diagnostic phase, it is easier to calculate dosimetry when using Lu-177. Conversely, Y-90 may permit the acquisition of low-quality images through Bremsstrahlung, a PET/CT acquisition also being possible, but at a high dosage, i.e., not in the diagnostic phase.

Consequently, waiting for the best results, which should be obtained using a tailored approach in each individual patient, a pivotal improvement in the clinical application of the theragnostic model based on SSA radio-chelates has been obtained from the NETTER-1 experience [37], a multicenter trial with the participation of 41 centers from 8 countries worldwide. It has been demonstrated that Lu-177-oxodotreotide is clinically useful in treating of well-differentiated (G1 and G2), progressive, non-removable or metastatic, somatostatin receptor-positive GEP-NETs. On the basis of the consequent approval by national and international authorities, the commercially available product Lutathera© may therefore be clinically utilized in patients with advanced midgut neuroendocrine tumors.

## 4. Future Perspectives and Conclusions

As NETs are rare, a favorable evolution of the application of the theragnostic SSA model may be supported by further multicenter trials. An active study in an advanced experimental state is the NETTER-2 study, aiming to evaluate whether Lu-177-DOTATATE may prolong, with respect to the current strategies, progression-free survival (PFS) in fast-growing GEP-NETs (G2 and G3). Another active study is the COMPETE study, a multicenter phase III trial to evaluate efficacy and safety of PRRT with Lu-177-edotreotide (Lu-177-DOTATOC) compared to targeted molecular therapy with everolimus in patients with inoperable GEP-NETs.

It must be however emphasized that many NETs cannot be effectively cured with Lutathera©.

It is the case that the resolution of symptoms or a reduction in the size of the lesions may be obtained in many of the patients positive at the diagnostic phase, using PET or SPECT with radiolabeled SSA. However, a complete remission is only achieved in a minority of them, which is also because NETs are usually diagnosed at an advanced stage, often already metastatic. Nevertheless, PRRT can be considered useful also in case a more cost-effective response or a superior palliative outcome may be reached with respect to alternative strategies. Radiolabeled SSA may therefore be considered to support a better quality of life in hopeless cases.

As a greater therapeutic efficacy may possibly be obtained by a higher dose of the currently used radiopharmaceutical, a first solution is certainly linked to the increase of the T/NT ratio, more easily obtainable through the reduction of the dose to the background. In PRRT, the major limiting factor is connected with renal dosimetry. Having the capability of reducing the dose to the kidneys, as may happen using drugs acting on radioactive renal uptake, it is possible to increase the therapeutic dose in PRRT, avoiding unsustainable kidney damage [19].

Better results should also be obtained using an integrated strategy based on different pathophysiological premises. In this sense, a longer (or more sustainable) survival should be obtained using rigorous protocols also combining other approaches such as chemotherapy, radiotherapy, immunotherapy, invasive therapeutic techniques, surgery and/or radioguided surgery [18,40], also defining a correct temporal sequence between the various procedures, i.e., trying to personalize the treatment. as much as possible.

However, if we want “to heal all patients with NET” we must be able to destroy all neoplastic cells, hopefully also in the case of dedifferentiation. To further improve therapeutic results in the group of patients with pathological conditions that already benefit from PRRT and/or to enlarge the ranks of subjects in which it can be effective, it is therefore relevant to understand where the weaknesses of this strategy lie.

At first, it is important to remember that SSR hyper-expression is typically associated with a well-differentiated state. Therefore, an improvement can be obtained with an earlier diagnosis, identifying a larger number of patients with almost all small homogeneous lesions, which can be treated more easily than larger ones. In this way, we could also reduce the occurrence of the escape phenomenon, associated with dedifferentiation, the undifferentiated neoplasm being frequently incurable, generally appearing at a more advanced phase of disease. To reach this goal, significant progress may be obtained through an earlier, more sensitive although accurate, screening of patients suspicious for NETs, using either complementary instrumental procedures (such as dynamic CT angiography, functional MRI, invasive US) or a better definition of the bio-molecular state. In this scenario, progress may be also associated with advanced and more performing hybrid technologies, with main reference to full digital PET/CT and PET/MRI.

With respect to native somatostatin, SSAs actually used in clinical practice may actively and effectively interact not with all the five SSRs. Therefore, a possible improvement, particularly relevant in tumors negative to the commercially available DOTA radiotracers (used either for diagnosis and therapy), as frequently happens for insulinoma, could be obtained both with a SSA changing its receptor spectrum or using a different approach, not based on the somatostatin model. At the present, some proposals have been made, suggesting somatostatin agonists with a different affinity, also including peptides showing affinity for all five SSRs [16], somatostatin antagonists [24], molecules interacting with a greater number of targets other than somatostatin [41]. Nevertheless, the “DOTA-Ga-68/Lu-177” model remains the most applied and effective.

A (theoretical) minor alternative, utilizable in individual cases, could be associated with routes of administration other than the intravenous, as the intra-arterial. This approach, eventually associated with a micro-embolization using radionuclides or not, could be utilized, for example, in patients with multiple hepatic lesions, some of which are otherwise untreatable.

Another (at present theoretical) intriguing perspective to treat undifferentiated neoplasms could be derived from the capability to re-differentiate malignant lesions or through a clinical application of translational research having the capability to introduce the sodium/iodide symporter gene in malignant cells, as premise to a high uptake of I-131 as sodium iodide [42].

In this context of perspectives regarding only a limited group of subjects or too far in the future (and probably difficult to realize widely in the clinical practice), the next and possible revolution can be associated with alpha emitters.

Until some years ago, the therapeutic action of the corpuscular nuclear radiations (beta- and alpha) seemed to be determined almost exclusively by the maximum range of the particles in the living matter. In this sense, because of the so-called “crossfire effect”, it was considered possible to reach a detrimental effect not only at the level of the cells englobing the radionuclide, but also in those non-expressing the target (either neoplastic or normal), but close to the concentrating site. The space within which this destructive effect manifests is related to the tissue attenuation power and to the energy spectrum of the radionuclide. Concerning beta emitters, for example in water, an effect may be determined up to 1.9 mm for Lu-177 and 11.8 mm for Y-90. This means that, in the comparison, Y-90 may provide a stronger and wider local extension of the therapeutic action, but in the presence of a greater collateral damage on normal cells.

The release of energy by alpha radiation is much larger than that released by beta radiation, but in the presence of a detrimental effect that generally occurs in a very short path, in the order of angstroms up to microns. Therefore, the crossfire effect is almost absent for alpha emitters, and it was hypothesized that the therapeutic action could be limited almost exclusively to the concentrating cells. This means that for many years, the administration of alpha emitters was considered very risky, in the absence of a magic bullet that guaranteed the concentration in the tumor cells but not in the normal cells. Therefore, its use was almost exclusively limited to research, often using selective and super-selective administration routes, also including the intra-tumor administration.

More recently, starting from the nineties, a great revolution has been brought about by the evidence that a detrimental action subsequent to the radioactive administration of radionuclide emitting corpuscle radiations, may be also dependent on the so-called “bystander” and “abscopal” effects.

The bystander effect is determined in cells adjacent to those with a target of a toxic effect. Being therefore an effect non-univocally determined by radiations, it has been identified as an important integrative therapeutic effect either in external radiotherapy and in radionuclide therapy. The detrimental action, mainly active on neoplastic cells but also with a possible involvement of normal cells residing in that tissue context, may be determined either by a transmembrane passage of a killing factor, or through the introduction into the interstitial space of toxic agents, capable of producing damage to the cells allocated in that stroma. In this way, together with targeted neoplastic cells concentrating the radionuclide, non-irradiated cells can also undergo a number of negative processes, ranging from an alteration of gene expression or in the process of translation, on cell multiplication, arriving to determine apoptosis and cellular death. Among the causes involved in determining the bystander effect we can consider epigenetic factors, some genes and molecules involved in the inflammation pathway, with free radicals being certainly involved and, in many cases, immunological determinants. Interestingly, this effect was experimentally demonstrated in 1992 by Nagasawa, using alpha particles. Although only 1% of cells were irradiated, a chromatid alteration was observed in more than 30% of the adjacent cells [43].

In 1953, R.H. Mole used the term “abscopal” (‘ab’—away from, ‘scopus’—target) to explain the effect of radiotherapy far from the irradiated volume [44]. It was hypothesized that the shrinkage of a treated neoplasm may produce some reactive effects by the organism able to destroy also distant metastases, out of the field of irradiation. In 2004, Demaria suggested that these anti-tumor effects were associated with the immune system [45]. It is evident that an abscopal effect should also occur in the case of a shrinkage of the tumor mass obtained with agents other than external radiotherapy, also regardless of radiations. In this context, the very high local damaging power determined by alpha particles can certainly activate an effective remote therapeutic action, even on neoplastic lesions that do not express the concentration’s target.

The mandatory premise of this favorable action is the availability of an “almost magical bullet” capable of concentrating with a more extreme focus on the pathological target than on normal tissue and emunctory systems.

The clear clinical evidence of this capability, i.e., of the relevance of the bystander and abscopal effects in determining a therapeutic efficacy of alpha emitters may be found in the clinical experience based on the use of Ra-223 in patients with skeletal metastases from prostate cancer. Better results having been demonstrated with respect to traditional strategies, such as palliative effects, a prognostic improvement with a longer survival has also been reported, suggesting a favorable effect also on untargeted lesions.

For these reasons, we believe that a marked improvement in PRRT could be obtained by identifying and commercially producing radio-chelates labelled with alpha particles, which have the ability to concentrate in high doses in neoplastic cells hyper-expressing SSR, in the presence of a low concentration in normal cells and in the emunctory systems. This action could be very effective, especially if adopted from the early stages of the disease. Initial results strongly support this hope.

While awaiting a promising future we can already improve the present. A theragnostic model for PRRT can certainly take advantage of the optimization of procedures, both by using more performing tools and by better linking information that comes from molecular biology and pathology with diagnostic imaging methods well-coordinated with each other.

A further improvement may derive from a more precise pathophysiological analysis, better connecting evidence concerning diagnostic and therapeutic radio-chelates, with main reference to their temporal biodistribution and to the dose to the kidneys.

In this sense, the future for NET patients can start improving immediately.

## 5. Take-Home Messages

Theragnostics is a strategy, strictly combining diagnosis and therapy, which creates the conditions for an “a priori” definition of an effective therapeutic effect;The most applied and effective model remains “DOTA-Ga-68/Lu-177”, used with success mainly in neuroendocrine tumors;More recently, other therapeutic radionuclides have been considered as therapeutic agents, with a growing interest in alpha emitters;“Bystander” and “Abscopal” effects could represent additional elements of therapeutic efficacy, especially with alpha emitters as demonstrated by the Ra-223 experience in patients with skeletal metastases from prostate cancer.

## Figures and Tables

**Figure 1 cancers-13-03474-f001:**
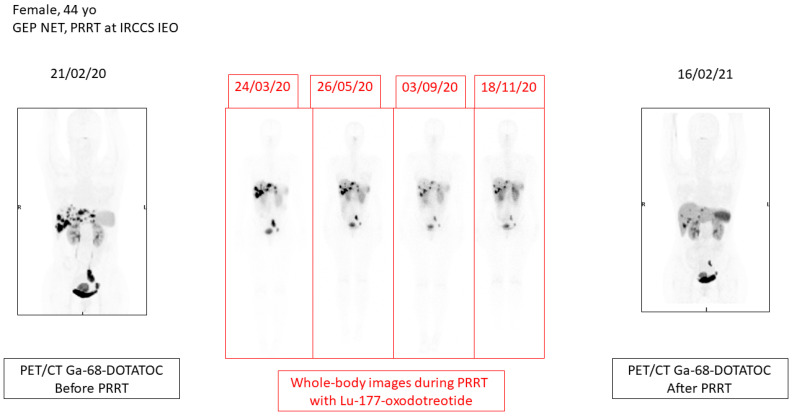
44-year-old female with GEP-NET who underwent a four-cycle treatment with Lu-177-oxodotreotide at IRCCS IEO. On the left, Ga-68-DOTATOC PET/CT before PRRT shows multiple foci of tracer uptake in the liver and abdomino-pelvic cavity. In the middle, four whole-body scans performed after every therapeutic dose of Lu-177-oxodotreotide, at 8–12 week time intervals. On the right, Ga-68-DOTATOC PET/CT after the last cycle of PRRT, which shows a significant reduction in disease burden with partial response to treatment.

**Table 1 cancers-13-03474-t001:** Principal theragnostic couples currently in use and investigational.

Theragnostic Molecules	Radiotracer Couples	Indication	Advantages	Limitations
SST2 agonists	^68^Ga-DOTATOC ^68^Ga-DOTANOC ^68^Ga-DOTATATE ^177^Lu-DOTATATE ^177^Lu-DOTATOC ^90^Y-DOTATOC	NETs, especially if GEP	Established imaging technique Approved for therapy (NETTER-1 trial) ^177^Lu potential labeling with non-carrier added ^90^Y causes high cross-fire effect	Renal/myelotoxicity
SST2 antagonists	^68^Ga-OPS201 ^68^Ga-OPS202 ^177^Lu-OPS201	NETs, especially if GEP	Superior sensitivity and tumor uptake compared to ^68^Ga/^177^Lu-DOTA-peptides with less renal toxicity	Myelotoxicity Still investigational
Iodine	^123^I ^131^I	Differentiated Thyroid Cancer (DTC)	Established imaging technique Approved for therapy of DTC and hyperthyroidisms	Myelotoxicity Long half-life of ^131^I
Metaiodobenzylguanidine	^123^I-MIBG ^131^I-MIBG	Pheochromocytoma and medullary thyroid carcinoma	Symptomatic relief in functioning tumors Absence of nephro-/hepatotoxicity	Myelotoxicity Limited treatment efficacy Thyroid blockage required
PSMA ligands	^68^Ga-PSMA-11 ^68^Ga-PSMA-617 ^177^Lu-PSMA-617	Metastatic prostate cancer	TheraP and VISION trials Possible future therapeutic option for metastatic castrate-resistant prostate cancer	Renal toxicity Prostate cancers not expressing PSMA

**Table 2 cancers-13-03474-t002:** Physical characteristics of some key radionuclides proposed for PRRT.

Radionuclide	Half-Life (HL)	Emissions	Mean Energy (keV)	Maximum Tissue Penetration
Indium-111	2.81 days	Conversion electrons	245	550 µm
		Auger electrons	25	10 µm
		Gamma rays	171, 245	
Yttrium-90	2.67 days	Beta- particles	933	12 mm
Lutetium-177	6.71 days	Beta- particles Gamma rays	133 208	2 mm
Iodine-131	8.0 days	Beta- particles Gamma rays	190 364	2.3 mm

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
