# Peer review of "From the Magic Bullet to Theragnostics: Certitudes and Hypotheses, Trying to Optimize the Somatostatin Model"

_cancers, 2021, doi:10.3390/cancers13143474_

Round 1

Reviewer 1 Report

This is a clear and in-depth review of the concept of theragnosis and its development, implantation and possible improvements in the models of somatostatin analogues.

Author Response

We would like to thank the reviewers for the constructive criticisms that have allowed us to improve the quality of our contribution.

Reviewer 2 Report

# Consider including NETs in the list of keywords;

# Consider adjusting the size of “Advantages” column (Table1) so that it does not require hyphenation of NETTER 1 Trial;

# On page 7 lines 337 and 353 it is written by Mansi et al and references 21 and 18 are mentioned. The first author in both cases is Cuccurullo V. We suggest the due correction;

# On page 13 it is written: “As it can be evidenced from the table ...... “

It should be clear which table is being referred to;

# Please, revise the references so that they are presented according to the rules for authors. Reference 6 is incomplete and references 7 and 8 should be corrected because the authors’ names are in capital letters.

Author Response

# Consider including NETs in the list of keywords;

NETs has been added to list of keywords.

# Consider adjusting the size of “Advantages” column (Table1) so that it does not require hyphenation of NETTER 1 Trial;

Sizing of Table 1 has been adjusted.

# On page 7 lines 337 and 353 it is written by Mansi et al and references 21 and 18 are mentioned. The first author in both cases is Cuccurullo V. We suggest the due correction;

Due corrections of cited references have been done.

# On page 13 it is written: “As it can be evidenced from the table ...... “

It should be clear which table is being referred to;

A table (Table 2) indicating key radionuclides involved for PRRT with their principal characteristics has been added.

# Please, revise the references so that they are presented according to the rules for authors. Reference 6 is incomplete and references 7 and 8 should be corrected because the authors’ names are in capital letters.

References have been revised accordingly.

Reviewer 3 Report

In the manuscript ”From the magic bullet to Theragnostics: certitudes and hypotheses, trying to optimize the somatostatin model”, the authors analyse the theranostic experience with somatostatin in neuroendocrine tumours and discuss the possibility of improving the current clinical results by extending the survival of a wider number of patients and by expanding the recruitment criteria to other pathologies.

The topic of this manuscript is very interesting and relevant to the scientific community and provides a significant contribution to the knowledge in the field. However, the clarity and the readability of the scientific text are at times blurred by inappropriate/unfamiliar choice of words as well as the use of very long and complicated sentences. I suggest a review of the manuscript since the text needs some modifications/corrections in order to be published

Find below some suggestions:

Simple Summary/Abstract: All the considerations made by the authors are relevant and scientifically correct. However, the text should be more straightforward and concise in order to disclose better the aim of the paper.

Introduction: Although the introduction contextualizes well the theme of the paper and guides the reader´s thoughts to the aim of the study it is too long and needs to be better organized. Some statements repeat themselves along the manuscript making the text needlessly long. Also, it is not very clear to me why the section 1.1 belongs to the Introduction. The title of section 1.1 is not very clear as well; what do the authors mean by “individuate in vitro a specific target”?

Section 2, Section 3 and Section 4. The text in the remaining sections suffers from the same drawbacks already pointed out: long text, unclear sentences, redundancies and ignorance of certain grammatical rules making it difficult to read and/or understand.

In brief, although the manuscript is scientific sound and interesting to warrant publication, it suffers from some drawbacks related to language and writing. The manuscript has numerous grammatical mistakes and redundancies, and the meaning of some sentences is not very clear. Please check throughout the text. As a non-native English speaker, I am well aware of how demanding it is to write in a foreign language, and therefore, I strongly suggest that the manuscript should be edited carefully by a native English speaker or someone with proficiency in English.

Author Response

At first, we would like to thank the reviewers for the constructive criticisms that have allowed us to improve the quality of our contribution.

In the manuscript ”From the magic bullet to Theragnostics: certitudes and hypotheses, trying to optimize the somatostatin model”, the authors analyse the theranostic experience with somatostatin in neuroendocrine tumours and discuss the possibility of improving the current clinical results by extending the survival of a wider number of patients and by expanding the recruitment criteria to other pathologies.

The topic of this manuscript is very interesting and relevant to the scientific community and provides a significant contribution to the knowledge in the field. However, the clarity and the readability of the scientific text are at times blurred by inappropriate/unfamiliar choice of words as well as the use of very long and complicated sentences. I suggest a review of the manuscript since the text needs some modifications/corrections in order to be published

Find below some suggestions:

Simple Summary/Abstract: All the considerations made by the authors are relevant and scientifically correct. However, the text should be more straightforward and concise in order to disclose better the aim of the paper.

Done.

Introduction: Although the introduction contextualizes well the theme of the paper and guides the reader´s thoughts to the aim of the study it is too long and needs to be better organized. Some statements repeat themselves along the manuscript making the text needlessly long. Also, it is not very clear to me why the section 1.1 belongs to the Introduction. The title of section 1.1 is not very clear as well; what do the authors mean by “individuate in vitro a specific target”?

The title of section 1.1 has been cancelled and the whole paragraph has been moved out of the introduction to Section 2.

Section 2, Section 3 and Section 4. The text in the remaining sections suffers from the same drawbacks already pointed out: long text, unclear sentences, redundancies and ignorance of certain grammatical rules making it difficult to read and/or understand.

In brief, although the manuscript is scientific sound and interesting to warrant publication, it suffers from some drawbacks related to language and writing. The manuscript has numerous grammatical mistakes and redundancies, and the meaning of some sentences is not very clear. Please check throughout the text. As a non-native English speaker, I am well aware of how demanding it is to write in a foreign language, and therefore, I strongly suggest that the manuscript should be edited carefully by a native English speaker or someone with proficiency in English.

The whole text has been revised and partially rewritten by a native English speaker. The presence of some repetitions is desired, because an answer has been given to the individual points addressed, sometimes taking up concepts already anticipated and resubmitted in a different way, in order to make the answer clearer.

This manuscript is a resubmission of an earlier submission. The following is a list of the peer review reports and author responses from that submission.